# Synergistic effect of collagen and CXCL12 in the low doses on human platelet activation

**Daiki Nakashima[1,2], Takashi Onuma[1], Kumiko Tanabe[1], Yuko Kito[1], Kodai Uematsu[2,3], Daisuke Mizutani[2,3], Yukiko Enomoto[3], Masanori Tsujimoto[3], Tomoaki Doi[4], Rie Matsushima-Nishiwaki[2], Haruhiko Tokuda[2,5], Shinji Ogura[4], Toru Iwama[5], Osamu Kozawa[2]\*, Hiroki Iida[1]**

1 Department of Anesthesiology and Pain Medicine, Gifu University Graduate School of Medicine, Gifu, Japan, 2 Department of Pharmacology, Gifu University Graduate School of Medicine, Gifu, Japan, 3 Department of Neurosurgery, Gifu University Graduate School of Medicine, Gifu, Japan, 4 Department of Emergency and Disaster Medicine, Gifu University Graduate School of Medicine, Gifu, Japan, 5 Department of Clinical Laboratory/Medical Genome Center Biobank, National Center for Geriatrics and Gerontology, Obu, Aichi, Japan

\* okkasugai@yahoo.co.jp

**Data Availability Statement:** All relevant data are within the manuscript and its Supporting Information files.

## Abstract

CXCL12, also known as stromal cell-derived factor-1, is a chemokine classified into CXC families, which exerts its function by binding to specific receptors called CXCR4 and CXCR7. Human platelets express CXCR4 and CXCR7 on the plasma membrane. It has been reported that CXCL12 potentiates to induce platelet aggregation in cooperation with agonists including collagen. However, the precise roles and mechanisms of CXCL12 in human platelet activation are not fully elucidated. In the present study, we investigated the effect of simultaneous stimulation with low doses of collagen and CXCL12 on the activation of human platelets. The simultaneous stimulation with collagen and CXCL12 induced the secretion of platelet-derived growth factor (PDGF)-AB and the release of soluble CD40 ligand (sCD40L) from human platelets in addition to their aggregation, despite the fact that the simultaneous stimulation with thrombin receptor-activating peptide (TRAP) or adenosine diphosphate (ADP), and CXCL12 had little effects on the platelet aggregation. The agonist of Glycoprotein (GP) VI convulxin and CXCL12 also induced platelet aggregation synergistically. The monoclonal antibody against CXCR4 but not CXCR7 suppressed the platelet aggregation induced by simultaneous stimulation with collagen and CXCL12. The phosphorylation of p38 mitogen-activated protein kinase (MAPK), but not p44/p42 MAPK, was induced by the simultaneous stimulation. In addition, the simultaneous stimulation with collagen and CXCL12 induced the phosphorylation of HSP27 and the subsequent release of phosphorylated-HSP27 from human platelets. SB203580, a specific inhibitor of p38 MAPK, attenuated the platelet aggregation, the phosphorylation of p38 MAPK and HSP27, the PDGF-AB secretion, the sCD40L release and the phosphorylated-HSP27 release induced by the simultaneous stimulation with collagen and CXCL12. These results strongly suggest that collagen and CXCL12 in low doses synergistically act to induce PDGF-AB secretion, sCD40L release and phosphorylated-HSP27 release from activated human platelets via p38 MAPK activation.

**Funding:** This study was supported in part by the Research Funding for Longevity Sciences (28-9 and 19-21) from National Center for Geriatrics and Gerontology, Japan, https://www.ncgg.go.jp/english/index.html Funding acquisition: OK, HT The funders had no role in study design, data collection and analysis, decision to publish, or preparation of the manuscript.

**Competing interests:** The authors have declared that no competing interests exist.

## Introduction

CXCL12, also called stromal cell-derived factor-1, is a chemokine classified into CXC families, heparin-binding proteins that direct the movement of circulating leukocytes to sites of inflammation or injury [1]. CXCL12 acts through its specific receptors CXCR4 and CXCR7, which are G-protein coupled receptors [2]. CXCL12 is involved in essential physiological processes such as embryogenesis, hematopoiesis and angiogenesis [3,4], and acts on many migrating cells and tissues such as primordial germ cells, neurons, lymphocytes, endothelial precursor cells and hematopoietic stem cells [3,4]. In addition to its involvement in physiological processes, CXCL12 is also involved in the pathogenesis of diseases such as cancer progression, neurodegenerative disorders, inflammatory bowel disease, rheumatic disease and coronary artery disease [2,4]. In hematopoietic systems, megakaryocyte lineage cells from colony forming units-megakaryocytes to mature megakaryocytes express CXCR4, and the CXCL12-CXCR4 axis is required to insert mature megakaryocytes into the appropriate vascular niche, a crucial process in platelet biogenesis [5].

Platelets, which critically plays a hemostatic role, are derived from the cytoplastic extension of mature megakaryocytes. Once a vessel wall is injured, platelets are tethered to the subendothelial matrix via adhesive receptors such as Glycoprotein (GP) Ib/IX/V with the interaction of von Willebrand factor [6]. The exposure of subendothelial collagen caused by the disruption of endothelium triggers the accumulation and activation of platelets through GPVI and integrin α2β1, which exist on platelets, leading to stabilized platelet adhesion and subsequent thrombus formation [7]. In addition to the expression of CXCR4, human platelets store CXCL12 in the α-granule and release it upon activation [2]. Platelet-derived CXCL12 is known to modulate paracrine mechanisms such as chemotaxis, adhesion, proliferation and differentiation of nucleated cells including progenitor cells, and enhances their recruitment to sites of vascular and tissue injury, resulting in the promotion of repair [2]. CXCL12, a weak agonist of platelet aggregation by itself, reportedly enhances the response to low-dose ADP, epinephrine, serotonin and a threshold concentration of collagen [8,9]. In addition, it has also been reported that the blockade of CXCR4 reduces collagen-induced thrombus formation [9]. Thus, evidence supporting the function of CXCL12 in human platelet activation is accumulating, however, the precise roles and mechanism underlying the involvement of CXCL12 in human platelet activation have yet to be fully clarified.

Accompanying activation, platelets secrete various mediators such as platelet-derived growth factor-AB (PDGF-AB) from the α-granule and release soluble CD40 ligand (sCD40L) as an inflammatory mediator [10,11]. These mediators released from activated platelets further accelerate platelet activation [6,7,12]. PDGF-AB is known to be a powerful mitogenic growth factor that promotes atherosclerosis by mainly acting in connective tissue including vascular smooth muscle cells [13]. sCD40L is known to induce the inflammatory process in the endothelium [14], and is thought to be an important factor involved in atherosclerosis pathogenesis [6].

Heat shock proteins (HSPs) are molecular chaperones induced by environmental stresses like chemical, metabolic and pathophysiological stresses [15,16]. HSPs, classified as major classes named HSPH (HSP110), HSPC (HSP90), HSPA (HSP70), DNAJ (HSP40), HSPB (Small HSP), HSPD/E (HSP60/HSP10) and CCT (TRiC) [16], are basically recognized to facilitate the refolding of damaged proteins intracellularly [15,16]. HSP27, one of HSPB, is constitutively expressed in various cells containing human platelets [16,17]. Three serine residues (Ser-15, Ser-78 and Ser-82) are phosphorylated through post-translational modification in human HSP27 [17,18]. Once phosphorylated, HSP27 is transformed into a form of dimer or monomer from an aggregated form, resulting in its modulation as a chaperone [15,19–21]. In addition to

its role as a molecular chaperone, the involvement of HSP27 in cancer development, cardio-vascular diseases, and inflammatory responses has been reported [19,22–24]. Besides the intra-cellular roles, extracellular HSP27 reportedly acts as an inflammation regulator [25]. Regarding platelet HSP27 in humans, we previously reported that HSP27 is phosphorylated by ADP or collagen stimulation through p44/p42 mitogen-activated protein kinase (MAPK) [26–28]. We also reported that collagen-stimulated human platelets release HSP27 into plasma following its phosphorylation in patients with type 2 diabetes mellitus (DM) [29].

In the present study, we investigated the synergistic effect of simultaneous stimulation with low dose collagen and CXCL12 on human platelet activation. We show that collagen and CXCL12 in low doses synergistically act to induce PDGF-AB secretion, sCD40L release and phosphorylated-HSP27 release from activated human platelets via p38 MAPK activation.

## Materials and methods

### Materials

Collagen was purchased from Takeda Austria GmbH (Linz, Austria). Thrombin receptor-activating peptide (TRAP, H-Ser-Phe-Leu-Leu-Arg-Asn-Pro-Asn-Asp-Lys-Tyr-Glu-Pro-Phe-OH trifluoroacetate salt) was purchased from Bachem (Bubendorf, Switzerland). ADP was purchased from Merck-Sigma-Aldrich (Darmstadt, Germany). Recombinant CXCL12, mouse anti-CXCR4 monoclonal antibody and mouse anti-CXCR7 monoclonal antibody were purchased from R&D systems (Minneapolis, MN). Control IgG was purchased from Santa Cruz Biotechnology, Inc. (Santa Cruz, CA). Convulxin was purchased from Cayman Chemical (Ann Arbor, MI). SB203580 was purchased from Calbiochem-Novabiochem Co. (La Jolla, CA). Phospho-specific p38 MAPK antibodies, p38 MAPK antibodies, phospho-specific p44/p42 MAPK antibodies, p44/p42 MAPK antibodies, were purchased from Cell Signaling Technology (Danvers, MA). Phospho-specific HSP27 (Ser-78) antibodies were purchased from Enzo Life Sciences, Inc. (Farmingdale, NY). GAPDH antibodies and HSP27 antibodies were purchased from Santa Cruz Biotechnology, Inc. (Santa Cruz, CA). PDGF-AB enzyme-linked immunosorbent assay (ELISA) kit and sCD40L ELISA kit were obtained from R&D Systems, Inc. (Minneapolis, MN). Phosphorylated-HSP27 (Ser-78) ELISA kit was purchased from Enzo Life Sciences, Inc. (Farmingdale, NY). Other materials and chemicals were obtained from commercial sources. SB203580 was dissolved in dimethyl sulfoxide. The maximum concentration of dimethyl sulfoxide was 0.3%, which did not affect platelet aggregation, protein detection using Western blotting or ELISA for PDGF-AB, sCD40L and phosphorylated-HSP27.

### Preparation of platelets

Human blood was donated from healthy volunteers, and immediately added to a 1/10 volume of 3.8% sodium citrate. Platelet-rich plasma (PRP) was obtained by centrifugation at $155 \times g$ at room temperature for 12 min. Platelet-poor plasma (PPP) was obtained from the residual samples by centrifugation at $1,400 \times g$ at room temperature for 5 min. This study was approved by the Ethics Committee of Gifu University Graduate School of Medicine (Gifu, Japan). Written informed consent was obtained from all of the participants.

### Platelet aggregation

Platelet aggregation was measured by PA-200 aggregometer (Kowa Co. Ltd., Tokyo, Japan), which can analyze the size of platelet aggregates based on particle counting by laser scattering methods (small; 9–25 μm, medium; 25–50 μm and large; 50–70 μm). Preincubation of PRP was performed at 37˚C for 1 min with a stirring speed of 800 rpm. PRP was stimulated by

CXCL12 with collagen, TRAP, ADP or convulxin. The dose of each stimulator achieving a % transmittance of 10%-30% recorded by the aggregometer was adjusted individually. When indicated, PRP was pretreated with anti-CXCR4 monoclonal antibody, anti-CXCR7 monoclonal antibody, control IgG or SB203580 for 3 min, and then stimulated by CXCL12 and/or collagen. The platelet aggregation was monitored for 4 min. The percentage of transmittance of the isolated platelets was recorded as 0%, and that of the PPP was recorded as 100%.

## Protein preparation after stimulation

After the stimulation by CXCL12 and/or collagen, platelet aggregation was terminated by adding an ice-cold EDTA (10 mM). The mixture was centrifuged at $10,000 \times g$ at 4˚C for 2 min. The supernatant was obtained for ELISA and stored at -80˚C. The pellet was washed twice with phosphate-buffered saline (PBS) and then lysed by boiling in a lysis buffer 62.5 mM Tris-HCl, pH 6.8, 2% sodium dodecyl sulfate (SDS), 50 mM dithiothreitol and 10% glycerol for a Western blotting analysis.

## Western blotting

Western blot analysis was performed as described previously [30]. Briefly, SDS-polyacrylamide gel electrophoresis (PAGE) was performed as described by Laemmli in 10% or 12.5% polyacrylamide gel [31]. The proteins in the gel were transferred onto a PVDF membrane and blocked with 5% fat-free dry milk in PBS containing 0.1% Tween 20 (PBS-T; 10 mM $Na_2HPO_4$, 1.8 mM $KH_2PO_4$, pH 7.4, 137 mM NaCl, 2.7 mM KCl, 0.1% Tween 20) for 2 h, then incubated with the indicated primary antibodies. Peroxidase-labeled anti-rabbit IgG antibodies were used as secondary antibodies. The primary and secondary antibodies were diluted to optimal concentrations with 5% fat-free dry milk in PBS-T. The peroxidase activity on the PVDF membrane was visualized on X-ray film using an ECL Western blotting detection system (GE Healthcare, Buckinghamshire, UK) as described in the manufacturer's protocol. A densitometric analysis was performed using a scanner and imaging software program (Image J version 1.50; NIH, Bethesda, MD, USA). The levels of phosphorylation were calculated as follows: the background-subtracted intensity of each signal was normalized to the respective intensity of GAPDH and plotted as the fold increase compared with that of the control cells.

## ELISA for PDGF-AB, sCD40L and phosphorylated-HSP27

The levels of PDGF-AB, sCD40L and phosphorylated-HSP27 (Ser-78) in the supernatant of the conditioned mixture after platelet aggregation were determined using ELISA kits for PDGF-AB, sCD40L and phosphorylated-HSP27 (Ser-78) respectively in accordance with the manufacturer's instructions.

## Statistical analysis

The data were analyzed by the Mann-Whitney U test. A probability of <5% was considered to be statistically significant. The data were presented as the mean ± SEM.

## Results

### Effect of simultaneous stimulation with collagen and CXCL12 in low doses on human platelet aggregation

It has been reported that CXCL12 enhances the response of platelet aggregation to a threshold concentration of collagen [9]. We evaluated the dose-response curves of the synergistic effects of collagen and CXCL12 in human platelet aggregation. We examined the effect of the

simultaneous stimulation of various doses (0.05 μg/ml, 0.1 μg/ml and 0.2 μg/ml) of collagen with a fixed dose (10 ng/ml) of CXCL12 on the platelet aggregation. Whereas collagen at a dose of up to 0.1 μg/ml with CXCL12 hardly increased the transmittance or size ratio of large aggregates, collagen at a dose of 0.2 μg/ml with CXCL12 significantly induced platelet aggregation (Fig 1A). We further examined the effect of the simultaneous stimulation of various doses (1 ng/ml, 3 ng/ml and 10 ng/ml) of CXCL12 with a fixed dose (0.2 μg/ml) of collagen on the platelet aggregation. Whereas CXCL12 up to 3 ng/ml with collagen had little effect on the transmittance or size ratio of large aggregates, CXCL12 at a dose of 10 μg/ml with collagen significantly induced platelet aggregation (Fig 1B).

## Effect of simultaneous stimulation with TRAP or ADP, and CXCL12 in low doses on human platelet aggregation

We examined the effects of TRAP or ADP in their subthreshold doses with CXCL12 (10 ng/ml) on human platelet aggregation. We here used TRAP instead of thrombin since TRAP is a potent activator of thrombin receptors, protease-activated receptors (PARs) [32]. When the platelets were stimulated in combination with TRAP (7 μM) and CXCL12 compared with TRAP alone, the increase of transmittance or the size ratio of large aggregates was not detected whereas the size ratios of small and medium aggregates were increased (Fig 2A). In addition, when the platelets were stimulated in combination with ADP (0.5 μM) and CXCL12 compared with ADP alone, the increase of transmittance or the size ratio of large aggregates was not detected whereas the size ratios of small and medium aggregates were increased (Fig 2B).

## Effect of simultaneous stimulation with convulxin and CXCL12 in low doses on human platelet aggregation

To further define the receptor for collagen in the synergistic effect of collagen and CXCL12 on human platelet aggregation, we examined the platelet aggregation stimulated with a combination of CXCL12 and convulxin. Convulxin is a snake venom activator of GPVI but not integrin

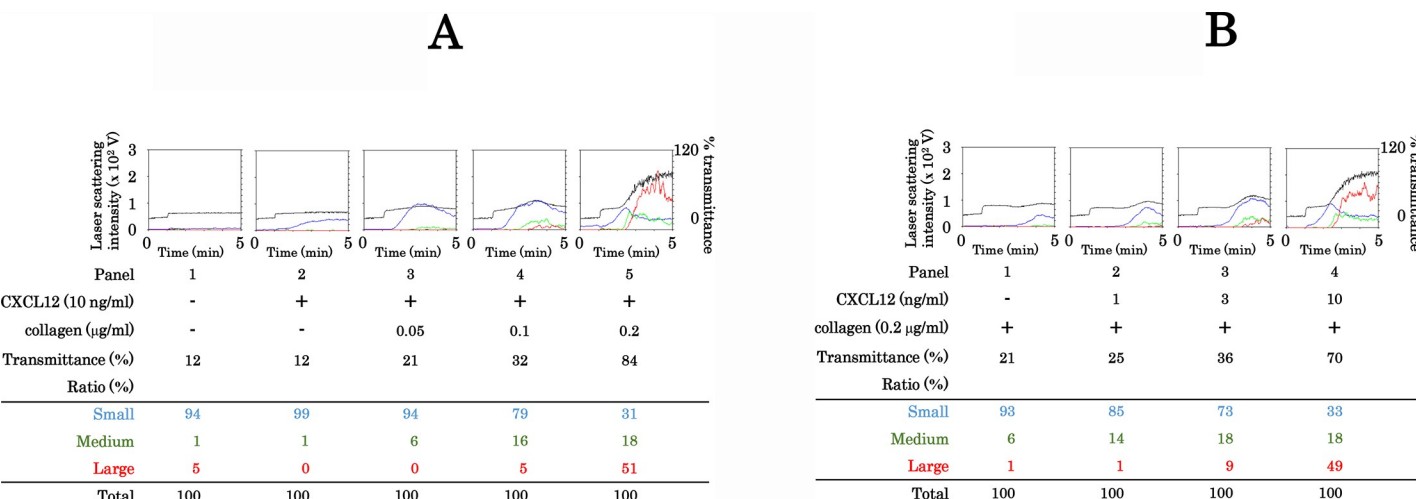

**A**

| Panel | 1 | 2 | 3 | 4 | 5 |
|---|---|---|---|---|---|
| CXCL12 (10 ng/ml) | - | + | + | + | + |
| collagen (μg/ml) | - | - | 0.05 | 0.1 | 0.2 |
| Transmittance (%) | 12 | 12 | 21 | 32 | 84 |
| Ratio (%) | | | | | |
| Small | 94 | 99 | 94 | 79 | 31 |
| Medium | 1 | 1 | 6 | 16 | 18 |
| Large | 5 | 0 | 0 | 5 | 51 |
| Total | 100 | 100 | 100 | 100 | 100 |

**B**

| Panel | 1 | 2 | 3 | 4 |
|---|---|---|---|---|
| CXCL12 (ng/ml) | - | 1 | 3 | 10 |
| collagen (0.2 μg/ml) | + | + | + | + |
| Transmittance (%) | 21 | 25 | 36 | 70 |
| Ratio (%) | | | | |
| Small | 93 | 85 | 73 | 33 |
| Medium | 6 | 14 | 18 | 18 |
| Large | 1 | 1 | 9 | 49 |
| Total | 100 | 100 | 100 | 100 |

**Fig 1. Effect of simultaneous stimulation with collagen and CXCL12 in low doses on platelet aggregation.** PRP was simultaneously stimulated by 0.05–0.2 μg/ml of collagen and 1–10 ng/ml of CXCL12 for 5 min. The reaction was terminated by the addition of ice-cold EDTA (10 mM) solution. The black line indicates the percentage of transmittance of each sample (isolated platelets recorded as 0%, and platelet-poor plasma recorded as 100%). The blue line indicates small aggregates (9–25 μm); green line, medium aggregates (25–50 μm); red line, large aggregates (50–70 μm). The lower panel presents the distribution (%) of aggregated particle size as measured by laser-scattering. The effects of simultaneous stimulation with 10 ng/ml of CXCL12 and 0.2 μg/ml or less of collagen (A), and simultaneous stimulation with 10 ng/ml or less of CXCL12 and 0.2 μg/ml of collagen (B) are shown.

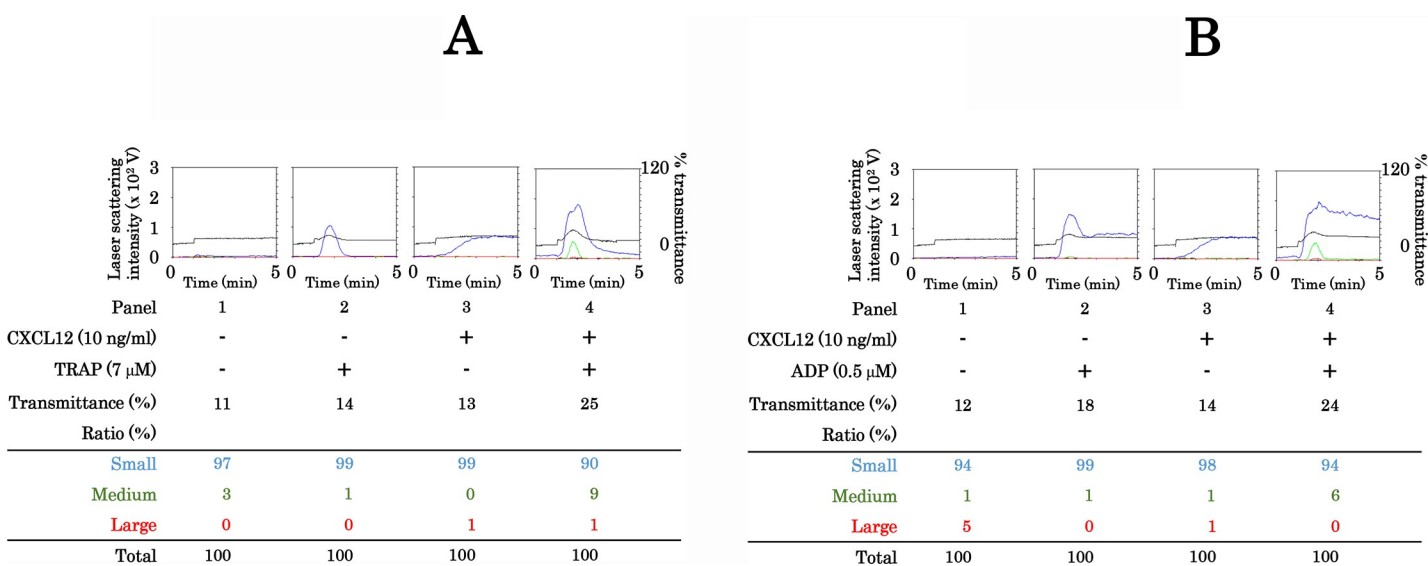

**Fig 2. Effect of simultaneous stimulation with TRAP or ADP, and CXCL12 in low doses on platelet aggregation.** PRP was simultaneously stimulated by 7 μM of TRAP (A) or 0.5 μM of ADP (B) and 10 ng/ml of CXCL12 for 5 min. The reaction was terminated by the addition of ice-cold EDTA (10 mM) solution. The black line indicates the percentage of transmittance of each sample (isolated platelets recorded as 0%, and platelet-poor plasma recorded as 100%). The blue line indicates small aggregates (9–25 μm); green line, medium aggregates (25–50 μm); red line, large aggregates (50–70 μm). The lower panel presents the distribution (%) of aggregated particle size as measured by laser-scattering. The representative results are shown.

α2/β3 [33]. Convulxin at a dose of 20 ng/ml had little effect on human platelet aggregation whereas 30 ng/ml of convulxin alone stimulated the aggregation. The combination of convulxin (20 ng/ml) and CXCL12 (10 ng/ml) synergistically stimulated the platelet aggregation (Fig 3).

### Effect of anti-CXCR4 or anti-CXCR7 monoclonal antibody on the human platelet aggregation induced by simultaneous stimulation with collagen and CXCL12

It is generally recognized that CXCL12 binds to both CXCR4 and CXCR7 receptors on human platelets [2]. Therefore, in order to define the receptor for CXCL12 in the synergistic effect of collagen and CXCL12 on the platelet aggregation, we examined whether anti-CXCR4 antibody and/or anti-CXCR7 antibody could neutralize the putative synergistic effect of CXCL12 with collagen on the platelet aggregation or not. The anti-CXCR4 antibody markedly suppressed the platelet aggregation stimulated by a combination of CXCL12 and collagen (Fig 4A). However, the synergistic effect of collagen and CXCL12 was hardly affected by anti-CXCR7 antibody (Fig 4B).

### Effect of simultaneous stimulation with collagen and CXCL12 in low doses on the secretion of PDGF-AB and release of sCD40L from human platelets

Collagen is known to induce the secretion of PDGF-AB and the release of sCD40L from activated human platelets [10,11]. We next examined the effect of simultaneous stimulation with collagen and CXCL12 in low doses on PDGF-AB secretion and sCD40L release from human platelets. Regarding the mechanism of sCD40L release from human platelets, CD40 ligand appears on the surface of activated platelets, and undergoes cleavage to release sCD40L [6]. On the other hand, PDGF-AB is secreted on time from α-granule accompanied with the platelet

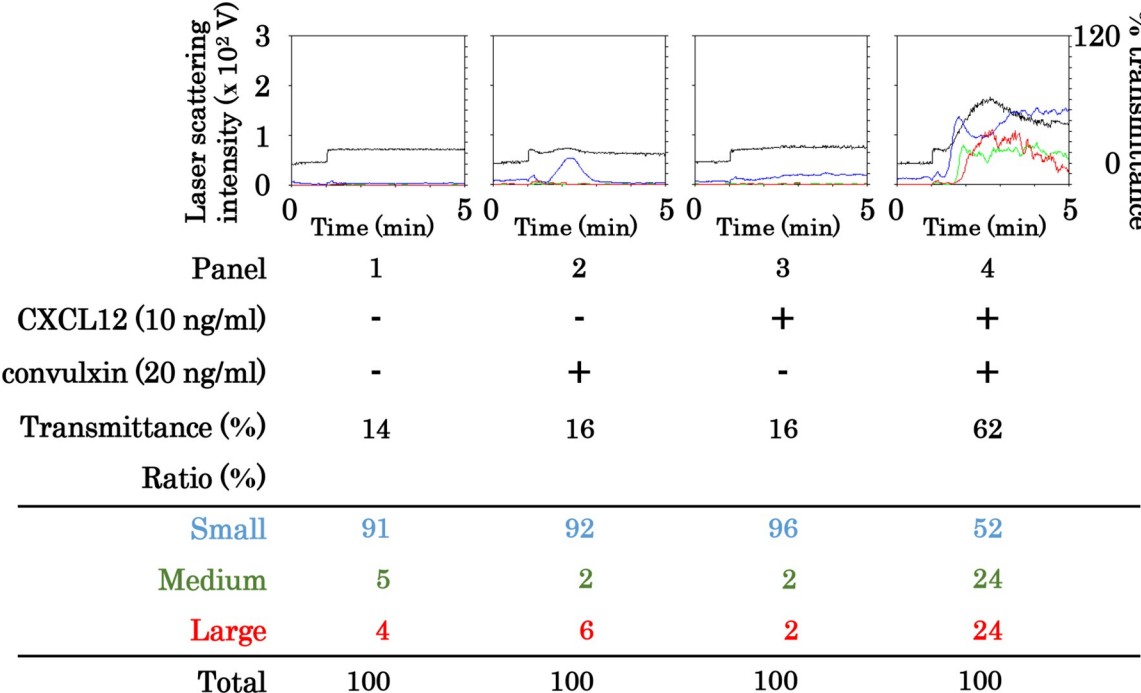

| | Panel | 1 | 2 | 3 | 4 |
|---|---|---|---|---|---|
| CXCL12 (10 ng/ml) | | - | - | + | + |
| convulxin (20 ng/ml) | | - | + | - | + |
| Transmittance (%) | | 14 | 16 | 16 | 62 |
| Ratio (%) | | | | | |
| Small | | 91 | 92 | 96 | 52 |
| Medium | | 5 | 2 | 2 | 24 |
| Large | | 4 | 6 | 2 | 24 |
| Total | | 100 | 100 | 100 | 100 |

**Fig 3. Effect of simultaneous stimulation with convulxin and CXCL12 in low doses on platelet aggregation.** PRP was simultaneously stimulated by 20 ng/ml of convulxin and 10 ng/ml of CXCL12 for 5 min. The reaction was terminated by the addition of ice-cold EDTA (10 mM) solution. The black line indicates the percentage of transmittance of each sample (isolated platelets recorded as 0%, and platelet-poor plasma recorded as 100%). The blue line indicates small aggregates (9–25 μm); green line, medium aggregates (25–50 μm); red line, large aggregates (50–70 μm). The lower panel presents the distribution (%) of aggregated particle size as measured by laser-scattering.

aggregation. We have previously reported that the release of sCD40L from activated platelets requires more than 5 min and reaches to the plateau up to 30 min after the stimulation [34]. Therefore, we adopted the co-stimulation lasted for 15 min in the release of sCD40L, whereas the co-stimulation for 5 min in the secretion of PDGF-AB. Collagen or CXCL12 in low doses could not solely induce PDGF-AB secretion, which is almost similar to vehicle. In contrast, simultaneous stimulation with collagen and CXCL12 markedly induced PDGF-AB secretion from human platelets (Fig 5). Regarding the effect on the release of sCD40L, low doses of collagen or CXCL12 did not induce sCD40L release alone, but their simultaneous stimulation with the same doses markedly induced sCD40L release from human platelets (Fig 6).

### Effect of simultaneous stimulation with collagen and CXCL12 in low doses on the phosphorylation of p44/p42 MAPK and p38 MAPK in human platelets

We previously reported that p44/p42 MAPK and/or p38 MAPK are involved in the PDGF-AB secretion and sCD40L release from activated human platelets [26,27]. Thus, we next examined the effects of simultaneous stimulation with collagen and CXCL12 in low doses on the phosphorylation of p44/p42 MAPK and p38 MAPK in human platelets. Low doses of collagen and CXCL12 both alone and in combination, failed to induce the phosphorylation of p44/p42 MAPK. (Fig 7). On the other hand, while low doses of collagen and CXCL12 alone failed to induce the phosphorylation of p38 MAPK, they markedly induced the phosphorylation of p38 MAPK in combination (Fig 8).

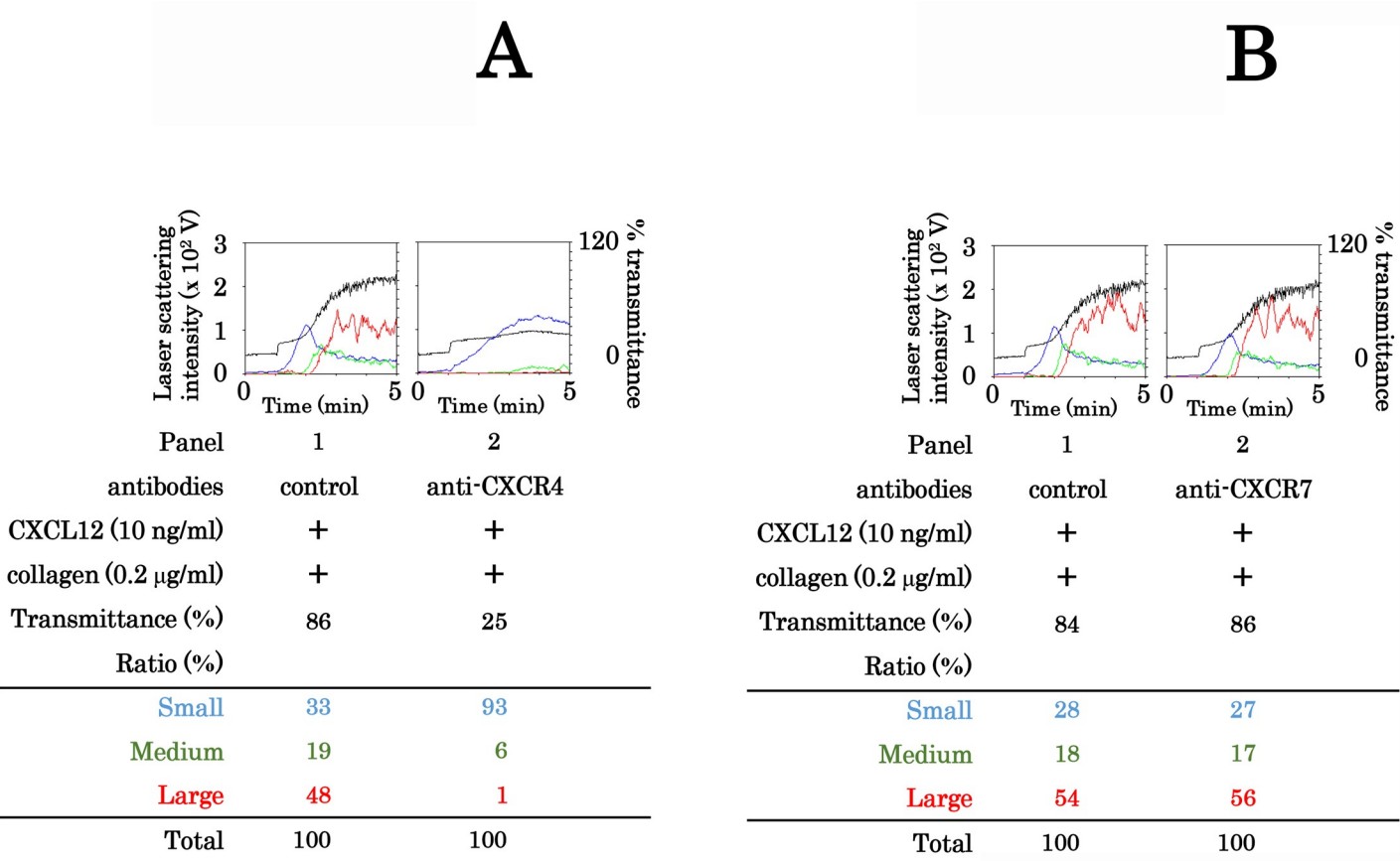

**Fig 4. Effect of anti-CXCR4 or anti-CXCR7 monoclonal antibody on the platelet aggregation induced by simultaneous stimulation with collagen and CXCL12 in low doses.** PRP was pretreated with 10 μg/ml of control IgG, anti-CXCR4 monoclonal antibody (A), or anti-CXCR7 monoclonal antibody (B) for 3 min and then simultaneously stimulated by 0.2 μg/ml of collagen and 10 ng/ml of CXCL12 for 5 min. The reaction was terminated by the addition of ice-cold EDTA solution. The black line indicates the percentage of transmittance of each sample (isolated platelets recorded as 0%, and platelet-poor plasma recorded as 100%). The blue line indicates small aggregates (9–25 μm); green line, medium aggregates (25–50 μm); red line, large aggregates (50–70 μm). The lower panel presents the distribution (%) of aggregated particle size as measured by laser-scattering.

### Effect of SB203580 on the human platelet aggregation induced by simultaneous stimulation with collagen and CXCL12

To investigate whether p38 MAPK is involved in the synergistic effect of collagen and CXCL12 on human platelet activation or not, we examined the effects of SB203580, a specific inhibitor of p38 MAPK [35], on the platelet aggregation induced by simultaneous stimulation with low dose collagen and CXCL12. The representative result is shown in Fig 9. SB203580 markedly suppressed the platelet aggregation that was synergistically induced by simultaneous stimulation with low dose collagen and CXCL12. Regarding the ratios of size, SB203580 markedly decreased the prevalence of medium aggregates (25–50 μm) and large aggregates (50–70 μm) but increased that of small aggregates (9–25 μm) (Table 1).

### Effect of SB203580 on the phosphorylation of p38 MAPK in human platelets induced by simultaneous stimulation with collagen and CXCL12 in low doses

We next examined the effect of SB203580 on the phosphorylation of p38 MAPK induced by simultaneous stimulation with low dose collagen and CXCL12 in human platelets. SB203580

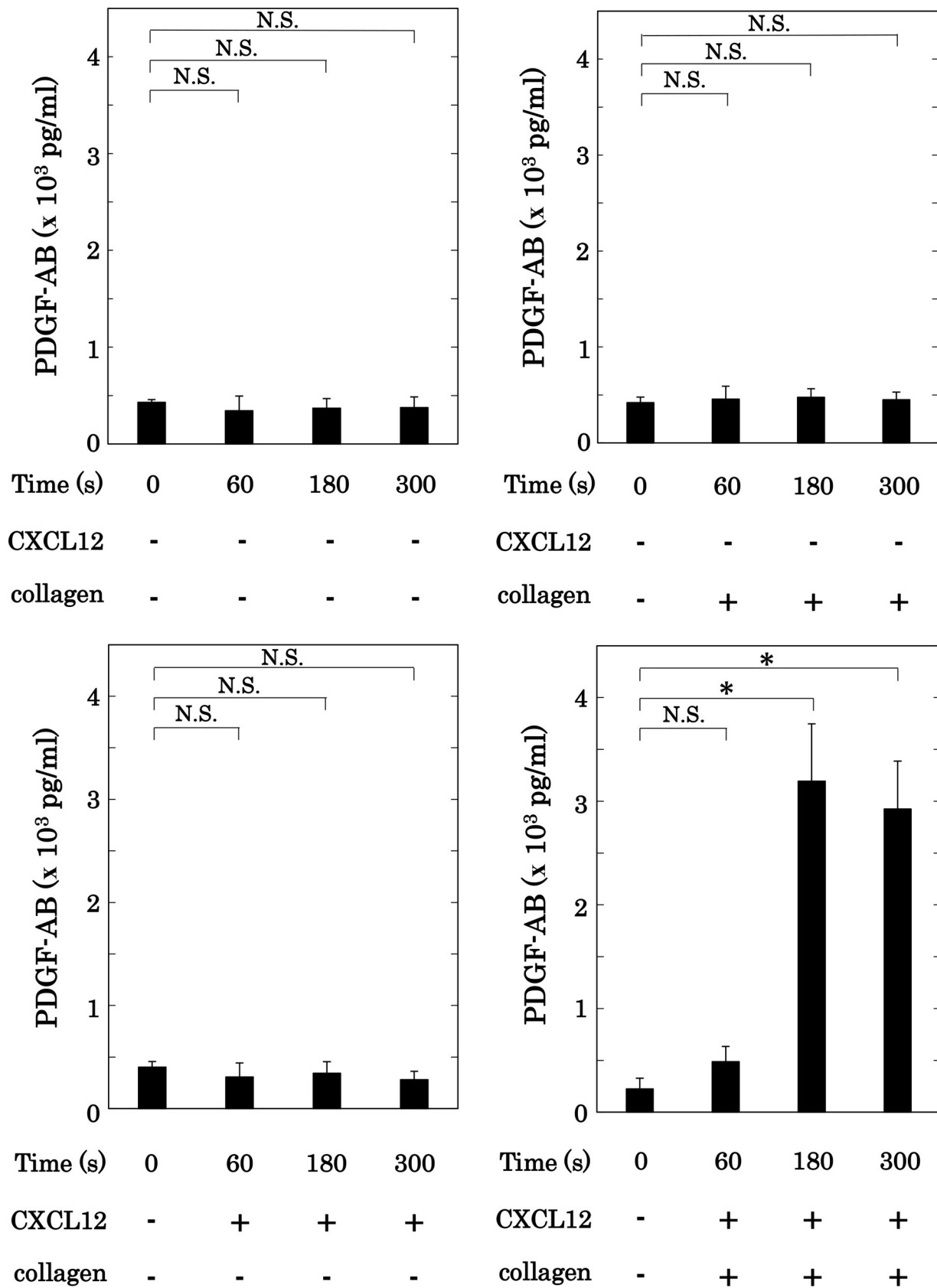

**Fig 5. Effect of simultaneous stimulation with collagen and CXCL12 in low doses on the secretion of PDGF-AB from human platelets.** PRP was simultaneously stimulated by the vehicles, 0.05–0.3 μg/ml of collagen and the vehicle, 10 ng/ml of CXCL12 and the vehicle, or 0.05–0.3 μg/ml of collagen and 10 ng/ml of CXCL12 for the indicated time. The dose of collagen achieving a % transmittance of 10%-30% was adjusted individually. The reaction was terminated by the addition of ice-cold EDTA (10 mM) solution. The conditioned mixture was centrifuged at 10,000 × g at 4°C for 2 min, and the supernatant was then subjected to ELISA for PDGF-AB. The results obtained from 5 individuals are shown. Each value represents the mean ± SEM. *p<0.05, compared to the value of control. NS: no significant differences between the indicated pairs.

(30 μM) significantly suppressed the phosphorylation levels of p38 MAPK simultaneously stimulated by collagen and CXCL12 at doses that alone had little effect on the phosphorylation (Fig 10).

### Effect of SB203580 on the secretion of PDGF-AB and release of sCD40L from human platelets induced by simultaneous stimulation with collagen and CXCL12 in low doses

We next examined the effects of SB203580 on the secretion of PDGF-AB and the release of sCD40L from human platelets simultaneously stimulated by low doses of collagen and

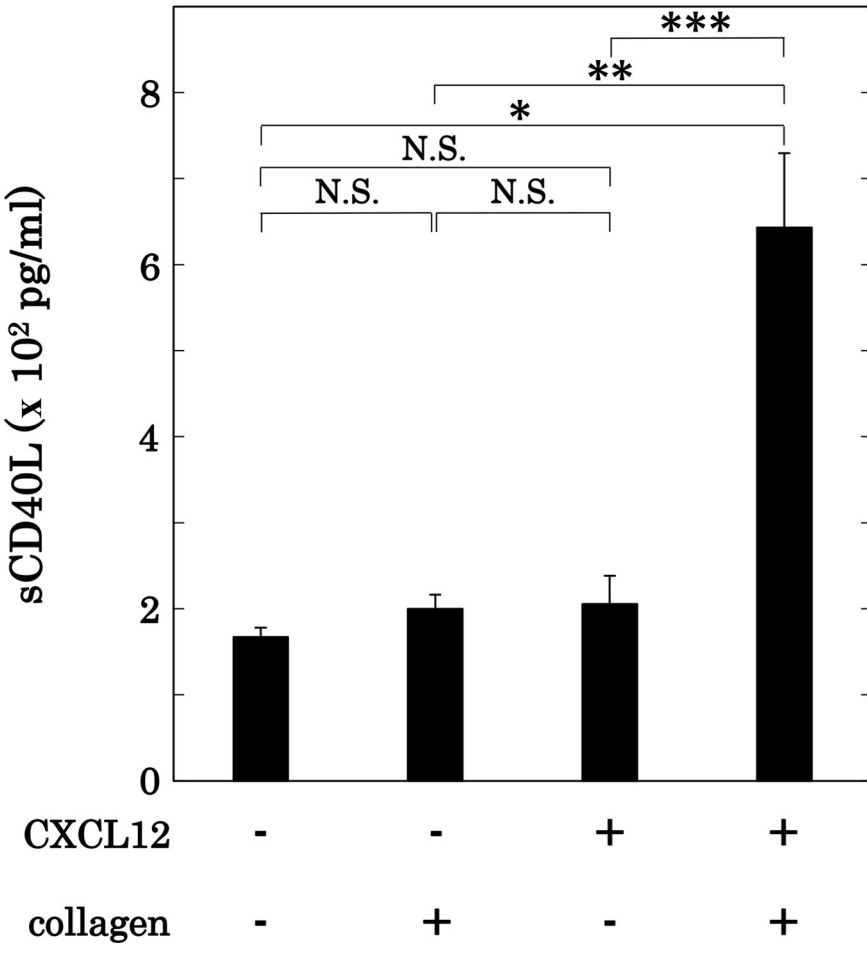

**Fig 6. Effect of simultaneous stimulation with collagen and CXCL12 in low doses on the release of sCD40L from human platelets.** PRP was simultaneously stimulated by 0.15–0.3 μg/ml of collagen and 10 ng/ml of CXCL12 for 15 min. The dose of collagen achieving a % transmittance of 10%-30% recorded was adjusted individually. The reaction was terminated by the addition of ice-cold EDTA (10 mM) solution. The conditioned mixture was centrifuged at 10,000 × g at 4°C for 2 min, and the supernatant was then subjected to ELISA for sCD40L. The results obtained from 5 independent individuals are shown. Each value represents the mean ± SEM. *p<0.05, compared to the value of control. **p<0.05, compared to the value of collagen alone. ***p<0.05, compared to the value of CXCL12 alone. NS: no significant differences between the indicated pairs.

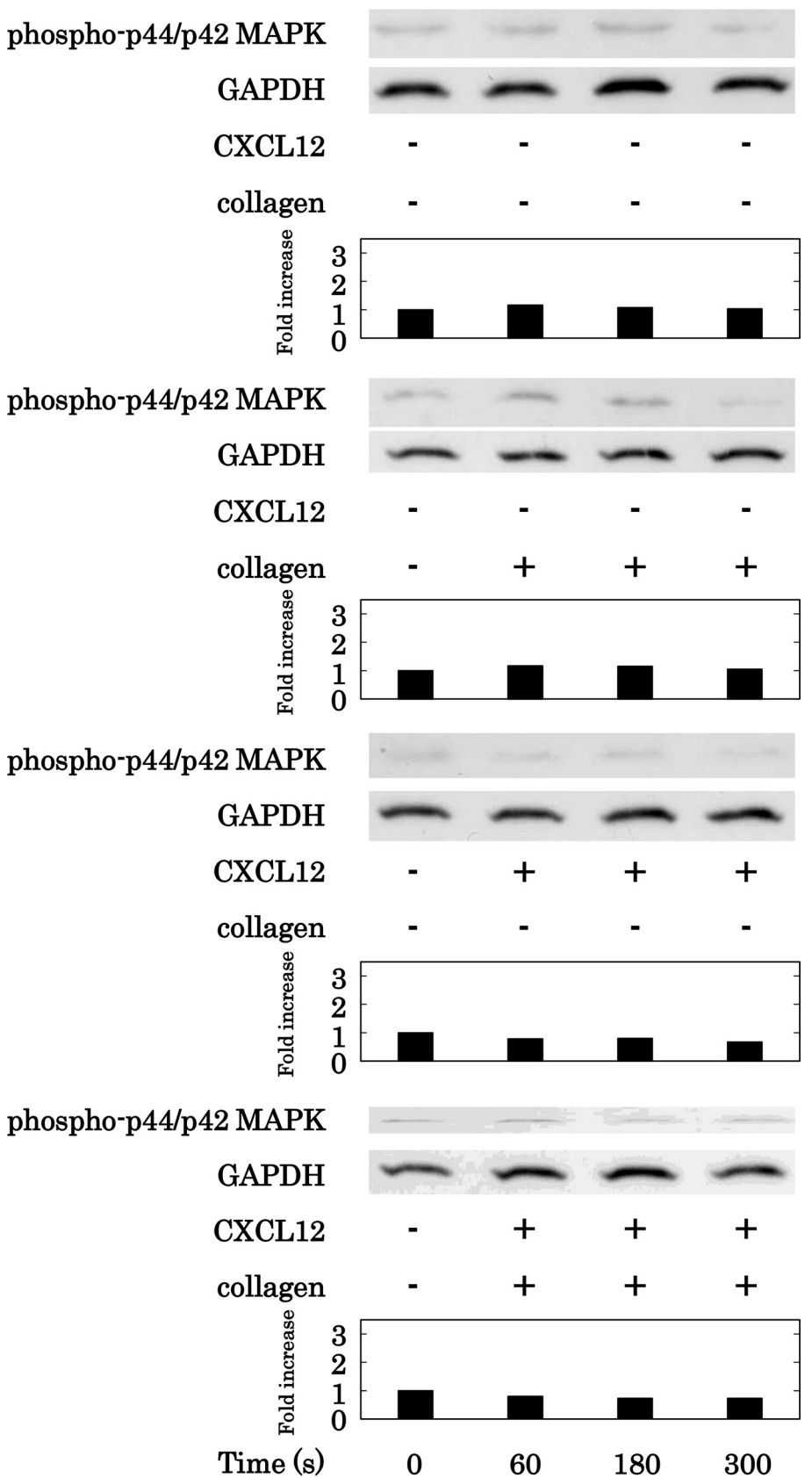

**Fig 7. Effect of simultaneous stimulation with collagen and CXCL12 in low doses on the phosphorylation of p44/p42 MAPK in human platelets.** PRP was simultaneously stimulated by 0.4 μg/ml of collagen and 10 ng/ml of CXCL12 for the indicated time. The reaction was terminated by the addition of ice-cold EDTA (10 mM) solution. The lysed platelets were subjected to Western blot analysis using antibodies against phospho-specific p44/p42 MAPK or GAPDH. The histogram shows a quantitative representation of the collagen and CXCL12-induced levels obtained from a densitometric analysis. The phosphorylation is expressed as the fold increase compared to the levels of platelets without stimulation. The representative results obtained from 5 independent individuals are shown.

CXCL12. SB203580 (30 μM) significantly reduced both the PDGF-AB secretion and the sCD40L release from human platelets synergistically stimulated by low doses of collagen and CXCL12 (Fig 11A and 11B).

## Effect of simultaneous stimulation with low dose collagen and CXCL12 on the phosphorylation of HSP27 and release of phosphorylated-HSP27 from human platelets

We showed that the synergistic effect of collagen and CXCL12 in low doses seems to involve the p38 MAPK pathway. It has been reported that p38 MAPK is involved in the phosphorylation of HSP27 induced by collagen in human platelets [36]. We therefore examined the effect of simultaneous stimulation with collagen and CXCL12 in low doses on the phosphorylation of HSP27 (Ser-78) in human platelets. The simultaneous stimulation with collagen and CXCL12 significantly increased the levels of HSP27 phosphorylation (Ser-78) at 180 s and 300 s (Fig 12A). In addition, we examined the effect of SB203580 on the phosphorylation of HSP27 (Ser-78) stimulated by a combination of collagen and CXCL12 in low doses in human platelets. SB203580 (30 μM) significantly suppressed the phosphorylation levels of HSP27 (Ser-78) induced by the synergistic effect of simultaneous stimulation with low dose collagen and CXCL12 in human platelets (Fig 12B). Regarding HSP27, we previously reported that collagen-stimulated human platelets from type 2 DM patients release HSP27 into plasma following its phosphorylation [29]. Therefore, we further examined the effect of simultaneous stimulation with low dose collagen and CXCL12 on the release of phosphorylated-HSP27 from human platelets. The low doses of CXCL12 and collagen synergistically induced the release of phosphorylated-HSP27 from human platelets. In addition, SB203580 (30 μM) markedly attenuated the release of phosphorylated-HSP27 (Ser-78) from human platelets induced by the simultaneous stimulation with collagen and CXCL12 in low doses (Fig 12C).

## Discussion

In the present study, we investigated the effect of simultaneous stimulation with subthreshold concentrations of collagen and CXCL12 in human platelets. It is recognized that activated human platelets induce PDGF-AB secretion from α-granule and sCD40L release [10,11]. We showed that simultaneous stimulation with collagen and CXCL12 induced PDGF-AB secretion and the release of sCD40L. We confirmed here that the simultaneous stimulation with collagen and CXCL12, which alone had no effect on platelet aggregation, dramatically induced aggregation as assessed using an aggregometer with laser scattering. Regarding the size of the platelet aggregates, simultaneous stimulation with collagen and CXCL12 increased the ratio of large aggregates but decreased the ratio of small aggregates, clearly indicating that collagen and CXCL12 synergistically act to potentiate platelet aggregation. Therefore, it is most likely that collagen and CXCL12 synergistically act to induce platelet activation, resulting in the secretion of PDGF-AB and release of sCD40L. From the dose-response reactions of collagen and CXCL12, it is probable that the synergistic effect of collagen and CXCL12 in the combination of 0.2 μg/ml of collagen and 10 ng/ml of CXCL12 is prominent to induce the platelet

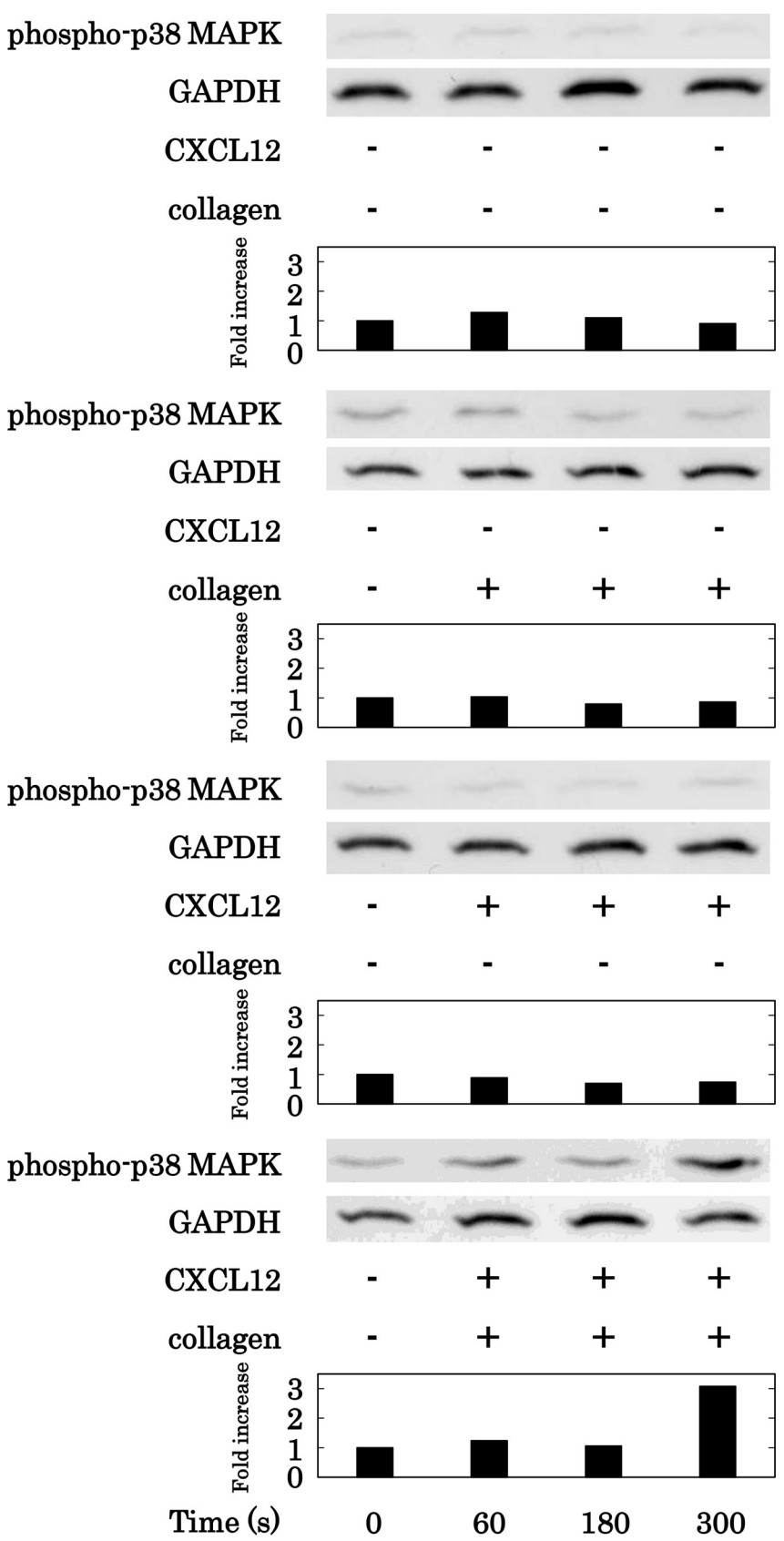

**Fig 8. Effect of simultaneous stimulation with collagen and CXCL12 in low doses on the phosphorylation of p38 MAPK in human platelets.** PRP was simultaneously stimulated by 0.4 μg/ml of collagen and 10 ng/ml of CXCL12 for the indicated time. The reaction was terminated by the addition of ice-cold EDTA (10 mM) solution. The lysed platelets were subjected to Western blot analysis using antibodies against phospho-specific p38 MAPK or GAPDH. The histogram shows a quantitative representation of the collagen and CXCL12-induced levels obtained from a densitometric analysis. The phosphorylation is expressed as the fold increase compared to the levels of platelets without stimulation. The representative results obtained from 5 independent individuals are shown.

aggregation, whereas the combination of these stimulators at lower doses is insufficient. The synergistic effects of thrombin or ADP with CXCL12 at subthreshold doses on platelet aggregation were hardly observed. Therefore, it is probable that the synergistic effect of CXCL12 on human platelet aggregation is specific to collagen. However, it has been reported that CXCL12 and TRAP as well as collagen synergistically stimulate platelet aggregation in mice [37]. The discrepancy between the present and previous findings might be due to differences in species or experimental conditions.

Regarding the exact mechanism underlying the synergistic effect of collagen and CXCL12 on human platelet aggregation, we showed that convulxin [33] mimicked the synergistic effect of collagen with CXCL12 on platelet aggregation, suggesting the involvement of GPVI as the receptor of collagen. Additionally, an antibody against CXCR4 but not CXCR7 suppressed the platelet aggregation induced by the simultaneous stimulation, suggesting the involvement of

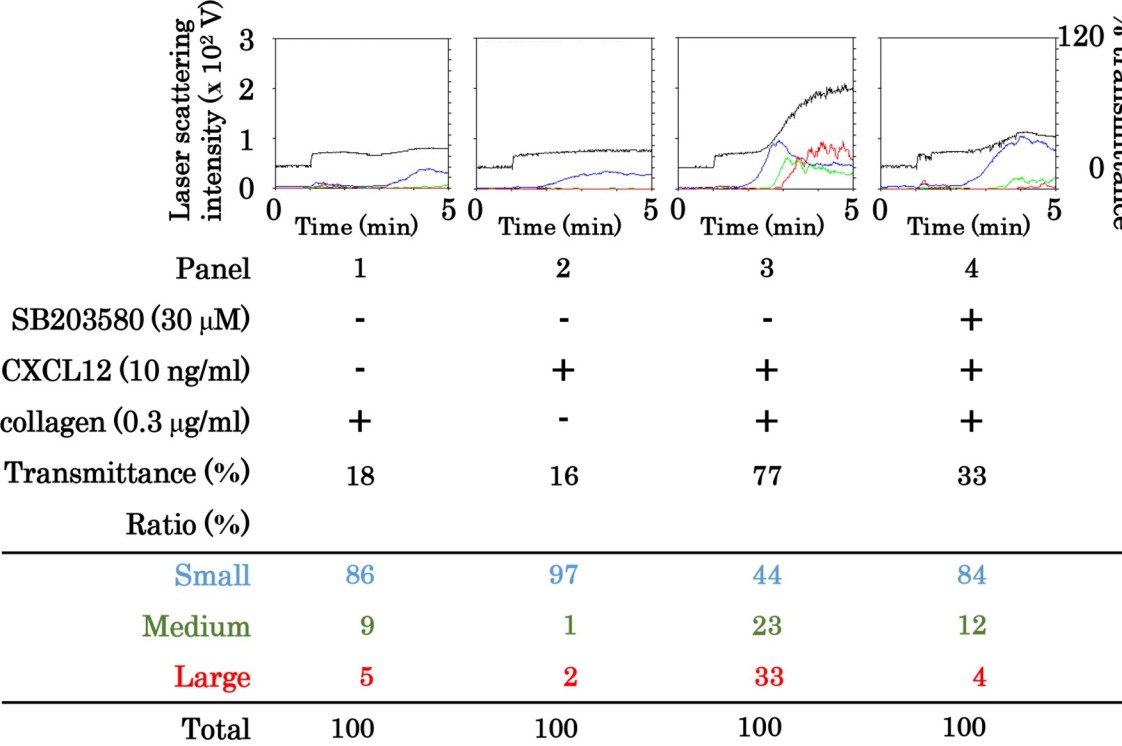

| Panel | 1 | 2 | 3 | 4 |
|---|---|---|---|---|
| SB203580 (30 μM) | - | - | - | + |
| CXCL12 (10 ng/ml) | - | + | + | + |
| collagen (0.3 μg/ml) | + | - | + | + |
| Transmittance (%) | 18 | 16 | 77 | 33 |
| Ratio (%) | | | | |
| Small | 86 | 97 | 44 | 84 |
| Medium | 9 | 1 | 23 | 12 |
| Large | 5 | 2 | 33 | 4 |
| Total | 100 | 100 | 100 | 100 |

**Fig 9. Effect of SB203580 on the platelet aggregation induced by simultaneous stimulation with collagen and CXCL12 in low doses.** PRP was pretreated with 30 μM of SB203580 or vehicle for 3 min and then simultaneously stimulated by 0.3 μg/ml of collagen and 10 ng/ml of CXCL12 for 5 min. The reaction was terminated by the addition of ice-cold EDTA solution. The black line indicates the percentage of transmittance of each sample (isolated platelets recorded as 0%, and platelet-poor plasma recorded as 100%). The blue line indicates small aggregates (9–25 μm); green line, medium aggregates (25–50 μm); red line, large aggregates (50–70 μm). The lower panel presents the distribution (%) of aggregated particle size as measured by laser-scattering. The representative results obtained from 5 healthy donors are shown.

**Table 1. Effect of SB203580 on platelet aggregation by low doses of collagen and CXCL12.**

| SB203580 | - | - | - | + |
|---|---|---|---|---|
| CXCL12 | - | + | + | + |
| collagen | + | - | + | + |
| Transmittance (%) | 14.0 ± 1.5 | 16.2 ± 1.9 | 68.4 ± 3.8* | 33.0 ± 2.7** |
| Large (%) | 7.0 ± 1.6 | 1.0 ± 0.5 | 30.0 ± 2.5* | 5.8 ± 0.6** |
| Medium (%) | 4.4 ± 0.5 | 1.0 ± 0.3 | 23.2 ± 0.4* | 14.0 ± 0.9** |
| Small (%) | 88.4 ± 1.9 | 98.0 ± 0.5 | 47.0 ± 2.6* | 80.0 ± 1.4** |

PRP was pretreated with 30 µM of SB203580 or vehicle for 3 min, and then simultaneously stimulated by collagen (0.2–0.3 µg/ml) and 10 ng/ml of CXCL12 for 5 min. The reaction was terminated by the addition of an ice-cold EDTA (10 mM) solution. The results obtained from the aggregometer for the transmittance and the ratio of large aggregates, medium aggregates and small aggregates, are summarized. Each value represents the mean ± SEM of 5 healthy donors.

*$p<0.05$, compared to the value of collagen or CXCL12 alone.

**$p<0.05$, compared to the value of collagen and CXCL12.

CXCR4 as the receptor for CXCL12. Based on our findings, it is most likely that the synergistic effect of CXCL12 and collagen on human platelet aggregation is mainly mediated through GPVI and CXCR4, the platelet membrane receptors for collagen and CXCL12, respectively.

We previously reported the involvement of p44/p42 MAPK and/or p38 MAPK in the PDGF-AB secretion and sCD40L release from activated human platelets [26,27]. Thus, we next investigated the engagement of p44/p42 MAPK and p38 MAPK in the synergistic effect of collagen and CXCL12 in low doses. We found that not p44/p42 MAPK but p38 MAPK was phosphorylated by the simultaneous stimulation with collagen and CXCL12. Therefore, it is likely that the activation of p38 MAPK could be involved in the synergistic effect of collagen and CXCL12 in human platelets. In order to confirm the role of p38 MAPK, we investigated the effect of SB203580 [35] on the synergistic effect of collagen and CXCL12 on the activation on human platelets, and found that SB203580 markedly attenuated the platelet aggregation, p38 MAPK phosphorylation, PDGF-AB secretion, and sCD40L release induced by the simultaneous stimulation of low dose collagen and CXCL12 in human platelets. The suppression by SB203580 of the effect of simultaneous stimulation with collagen and CXCL12 on the platelet activation was not complete, suggesting that pathway(s) in addition to p38 MAPK might be involved in the synergistic effect. It is most likely that the synergistic effect of simultaneous stimulation with collagen and CXCL12 on human platelet activation is mediated, at least in part, through the p38 MAPK pathway.

In our previous study [29], we showed that collagen-stimulated platelets release HSP27 into plasma following its phosphorylation in DM patients. It has been reported that p38 MAPK is involved in the collagen-induced HSP27 phosphorylation [29,36]. Based on these previous findings, we additionally investigated the effect of simultaneous stimulation with collagen and CXCL12 on the phosphorylation of HSP27 (ser-78) and the release of phosphorylated-HSP27 in human platelets. We found that simultaneous stimulation with low dose collagen and CXCL12 truly induced the phosphorylation of HSP27 and release of phosphorylated-HSP27. We also found that SB203580 markedly attenuated the phosphorylation of HSP27 and release of phosphorylated-HSP27 from human platelets induced by the simultaneous stimulation with low dose collagen and CXCL12. Therefore, it is probable that the simultaneous stimulation with collagen and CXCL12 synergistically induces the phosphorylation of HSP27 and subsequent release of phosphorylated HSP27 from activated human platelets, and the p38 MAPK is involved in the phosphorylation of HSP27 leading to the release. Taken together, our present findings strongly suggest that collagen and CXCL12 synergistically affect the platelet activation, leading to the aggregation, PDGF-AB secretion, sCD40L release and phosphorylation of

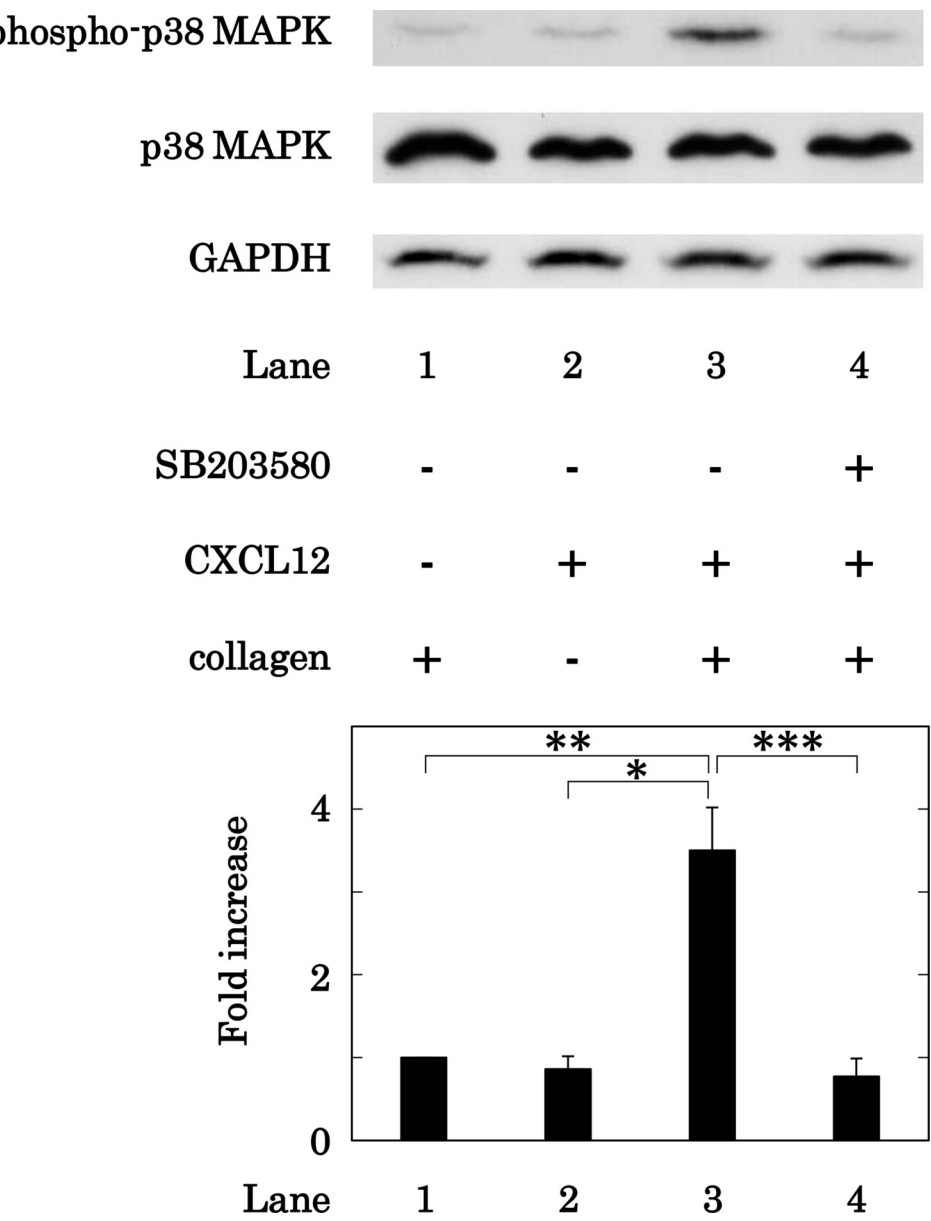

**Fig 10. Effect of SB203580 on the phosphorylation of p38 MAPK in human platelets induced by simultaneous stimulation with collagen and CXCL12 in low doses.** PRP was pretreated with 30 μM of SB203580 or vehicle for 3 min and then simultaneously stimulated by 0.1–0.2 μg/ml of collagen and 10 ng/ml of CXCL12 for 5 min. The reaction was terminated by the addition of ice-cold EDTA (10 mM) solution. The lysed platelets were subjected to Western blot analysis using antibodies against phospho-specific p38 MAPK, total p38 MAPK, or GAPDH. The histogram shows a quantitative representation of the co-stimulation with collagen and CXCL12-induced levels obtained from a densitometric analysis. The phosphorylation is expressed as the fold increase compared to the levels of collagen alone, presented as lane 1. Each value represents the mean ± SEM of 5 times independent experiments. $^{*}$p<0.05, compared to the value of CXCL12 alone. $^{**}$p<0.05, compared to the value of collagen alone. $^{***}$p<0.05, compared to the value of simultaneous stimulation of collagen and CXCL12.

HSP27 associated with the subsequent release into plasma, which is mediated, at least in part, through the p38 MAPK pathway.

It has been reported that CXCL12 is highly induced in some pro-atherogenic pathological conditions such as hyperlipidemia and diabetes [38,39]. The serum level of CXCL12 is

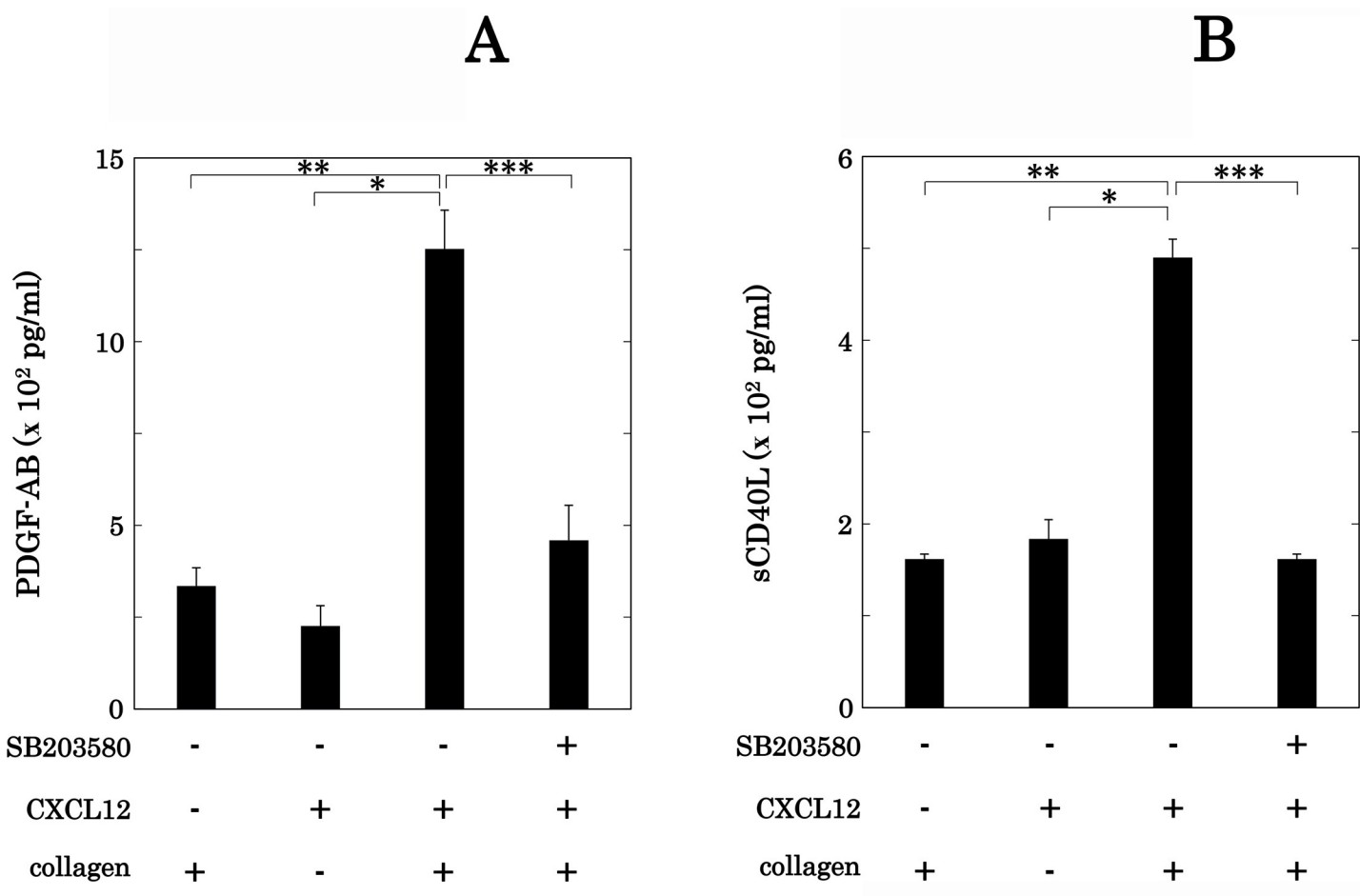

**Fig 11. Effects of SB203580 on the secretion of PDGF-AB and the release of sCD40L from human platelets induced by simultaneous stimulation with collagen and CXCL12 in low doses.** PRP was pretreated with 30 μM of SB203580 or vehicle for 3 min and then simultaneously stimulated by 0.1–0.2 μg/ml of collagen and 10 ng/ml of CXCL12 for 5 min (A) or 15 min (B). The dose of collagen achieving a % transmittance of 10%-30% recorded in an aggregometer was adjusted individually. The reaction was terminated by the addition of ice-cold EDTA (10 mM) solution. The conditioned mixture was centrifuged at $10,000 \times g$ at 4˚C for 2 min, and the supernatant was then subjected to ELISA for PDGF-AB (A) or sCD40L (B). The results from 5 independent individuals are shown. Each value of PDGF-AB (A) or sCD40L (B) represents the mean ± SEM. *$p < 0.05$, compared to the value of CXCL12 alone. **$p < 0.05$, compared to the value of collagen alone. ***$p < 0.05$, compared to the value of simultaneous stimulation of collagen and CXCL12.

reportedly increased in type 2 diabetic patients (healthy controls: 58 pg/ml, type 2 diabetic patients: 204.24 pg/ml) [40], which are lower than that used in our experiments. CXCL12 is also reported to be highly expressed in smooth muscle cells, endothelial cells and macrophages in human atherosclerotic plaques but not in normal vessels [41]. Therefore, under these pathological conditions, it is probable that platelets are easily activated by the combination of CXCL12 and subendothelial collagen at the site of vascular injury in chronic inflammatory atherosclerotic lesions, which could trigger thromboembolic diseases such as myocardial infarction and brain stroke. Indeed, increased plasma levels of CXCL12 reportedly could predict acute coronary diseases and future stroke [42–44]. The synergistic effect of collagen and CXCL12 on the platelet activation seems to be a mechanism underlying an increased risk of vascular diseases. Furthermore, the platelets synergistically activated by the combination of CXCL12 and collagen could secrete PDGF-AB and release sCD40L. It is well known that PDGF-AB acts as a powerful mitogenic growth factor that leads to atherosclerosis promotion by acting on connective tissue like vascular smooth muscle cells [13]. It is also known that

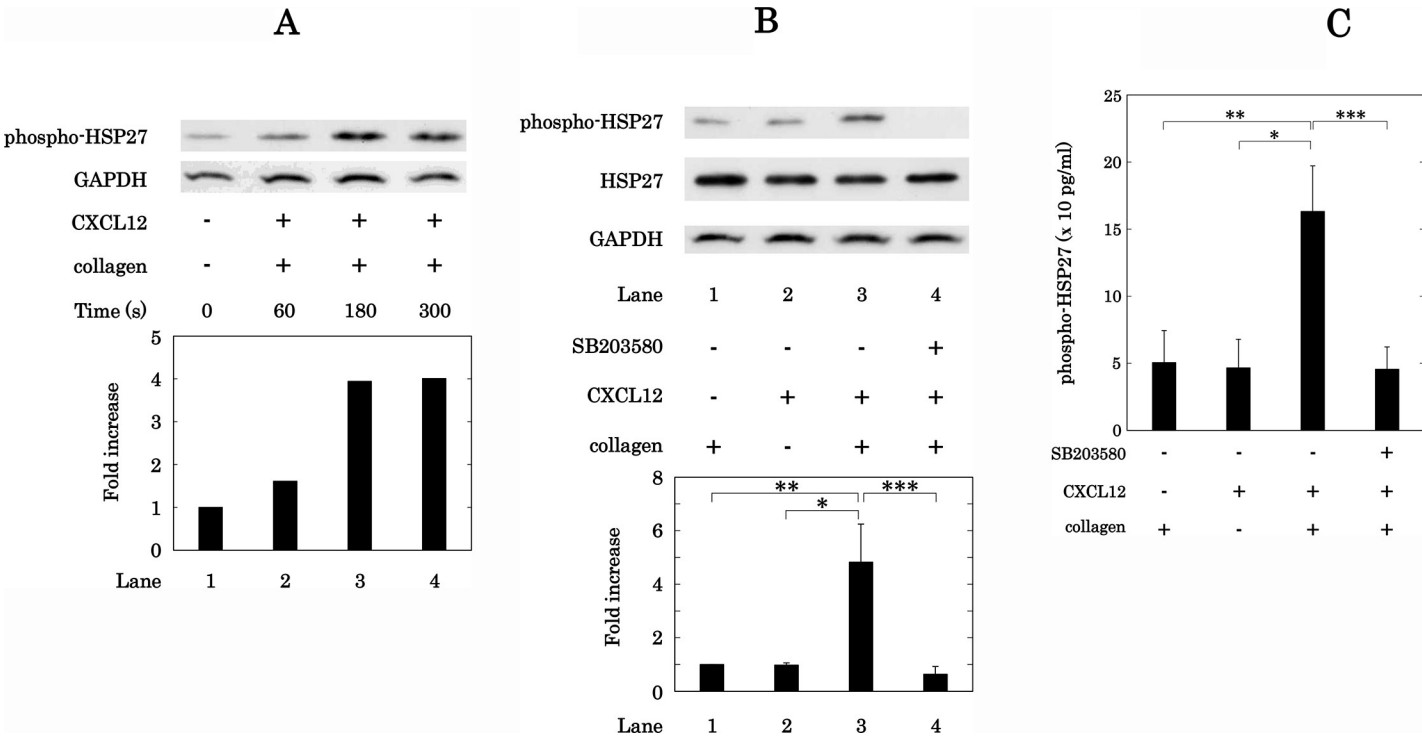

**Fig 12. Effects of simultaneous stimulation with collagen and CXCL12 in low doses on the phosphorylation of HSP27 and release of phosphorylated-HSP27 in human platelets.** PRP was simultaneously stimulated by 0.4 μg/ml of collagen and 10 ng/ml of CXCL12 for the indicated time. The reaction was terminated by the addition of ice-cold EDTA (10 mM) solution. The lysed platelets were subjected to Western blot analysis using antibodies against phospho-specific HSP27 (Ser-78) or GAPDH. The histogram shows a quantitative representation of the collagen and CXCL12-induced levels obtained from a densitometric analysis. The phosphorylation is expressed as the fold increase compared to the levels of platelets without stimulation. The representative result is shown (A). PRP was pretreated with 30 μM of SB203580 or vehicle for 3 min and then simultaneously stimulated by 0.1–0.2 μg/ml of collagen and 10 ng/ml of CXCL12 for 5 min. The reaction was terminated by addition of ice-cold EDTA (10 mM). The extracts of platelets were then subjected to SDS-PAGE with a subsequent Western blot analysis using antibodies against phospho-specific HSP27 (Ser-78), total HSP27, or GAPDH. The histogram shows a quantitative representation of the collagen and CXCL12-induced levels obtained from a densitometric analysis. The phosphorylation is expressed as the fold increase compared to the levels of collagen alone, presented as lane 1 (B). PRP was pretreated with 30 μM of SB203580 or vehicle for 3 min and then stimulated by 0.1–0.2 μg/ml of collagen and 10 ng/ml of CXCL12 for 15 min. The reaction was terminated by the addition of ice-cold EDTA solution. The mixture was centrifuged at $10,000 \times g$ at 4°C for 2 min, and the supernatant was then subjected to ELISA for phosphorylated-HSP27 (Ser-78) (C). Each value represents the mean ± SEM of the 5 times independent experiments. *$p < 0.05$ compared to the value of CXCL12 alone. **$p < 0.05$ compared to the value of collagen alone. ***$p < 0.05$ compared to the value of collagen and CXCL12 (B, C).

sCD40L induces inflammatory responses to the endothelium in humans [14]. Therefore, it is probable that these mediators from platelets synergistically activated by the combination of collagen and CXCL12 may contribute to the exacerbation of atherosclerotic pathology.

On the other hand, extracellular HSP27 is reported to function as an atheroprotective agent through the regulation of cholesterol homeostasis and modulation of inflammatory responses [23]. It has been reported that HSP27 functions to the nuclear factor κB activation in macrophages, relating to the secretion of both pro-inflammatory factors like IL-1β and anti-inflammatory factors like IL-10 [25]. Atheroprotective effect of extracellular HSP27 is assumed to be on the balance between anti-inflammatory and pro-inflammatory factors [25]. Additionally, HSP27 released from ischemic myocardium reportedly induces an inflammatory response in human coronary vascular endothelium cells, which is thought to be mediated through toll-like receptor (TLR) 2 and TLR 4 [45]. It is widely accepted that inflammation is closely related to atherosclerogenesis [6]. The increased extracellular HSP27 released from platelets synergistically activated by a combination of collagen and CXCL12 might also be involved in the progression of atherosclerosis. The progression of atherosclerotic pathology, therefore, might be also accelerated by the release of HSP27 from platelets, which could stimulate the vascular

smooth muscle cell mitogenesis and the inflammatory responses of vascular endothelium cells in cooperation with PDGF-AB and sCD40L. Taking our results into account as a whole, the synergistic effect of CXCL12 and collagen mediated, at least in part, via the p38 MAPK pathway might be one of therapeutic targets for the prevention of atheroprogression and thromboembolic disease in patients strongly expressing CXCL12. Further investigations will be needed in order to clarify the details.

In conclusion, our results strongly suggest that collagen and CXCL12 in low doses synergistically act to induce PDGF-AB secretion, sCD40L release and phosphorylated-HSP27 release from activated human platelets via p38 MAPK activation.

## Acknowledgments

We are very grateful to Mrs. Yumiko Kurokawa for her skillful technical assistance.

## Author Contributions

**Data curation:** Osamu Kozawa.

**Formal analysis:** Daiki Nakashima.

**Funding acquisition:** Haruhiko Tokuda, Osamu Kozawa.

**Investigation:** Daiki Nakashima, Kodai Uematsu, Daisuke Mizutani, Rie Matsushima-Nishiwaki.

**Methodology:** Osamu Kozawa.

**Project administration:** Osamu Kozawa.

**Resources:** Osamu Kozawa.

**Software:** Daiki Nakashima.

**Supervision:** Osamu Kozawa, Hiroki Iida.

**Validation:** Kumiko Tanabe, Yuko Kito, Yukiko Enomoto, Masanori Tsujimoto, Tomoaki Doi, Shinji Ogura, Toru Iwama, Osamu Kozawa.

**Visualization:** Daiki Nakashima, Kodai Uematsu.

**Writing – original draft:** Daiki Nakashima, Takashi Onuma, Haruhiko Tokuda, Osamu Kozawa.

**Writing – review & editing:** Daiki Nakashima, Takashi Onuma, Haruhiko Tokuda, Osamu Kozawa.

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
