## [Decision Letter · Decision Letter 0]

31 Mar 2020

PONE-D-19-34102

Synergistic effect of collagen and CXCL12 in the low doses on human platelet activation

PLOS ONE

Dear Dr. Kozawa,

Thank you for submitting your manuscript to PLOS ONE. After careful consideration, we feel that it has merit but does not fully meet PLOS ONE’s publication criteria as it currently stands. Therefore, we invite you to submit a revised version of the manuscript that addresses the points raised during the review process.

The manuscript reports the effect of collagen and the chemokine CXCL12 on platelet aggregation and release of growth factor PDGF-AB, cytokine sCD40L and phosphorylated HSP27. Although the hypothesis is interesting, this study remains preliminary and descriptive.  Further experiments on the type of platelet activation and its responsive receptors (ADP, thrombin, GPVI) should be performed, and dose-response curves of the synergistic effects of the platelet agonist collagen and CXCL12 should be evaluated. In addition, the specific points from the reviewers need a point to point response. 

We would appreciate receiving your revised manuscript by May 15, 2020. To enhance the reproducibility of your results, we recommend that if applicable you deposit your laboratory protocols in protocols.io, where a protocol can be assigned its own identifier (DOI) such that it can be cited independently in the future. For instructions see: http://journals.plos.org/plosone/s/submission-guidelines#loc-laboratory-protocols

We look forward to receiving your revised manuscript.

Kind regards,

Yong Jiang, Ph.D.

Academic Editor

PLOS ONE

Journal Requirements:

Reviewers' comments:

Reviewer's Responses to Questions

**Comments to the Author**

1. Is the manuscript technically sound, and do the data support the conclusions?

Reviewer #2: Partly

Reviewer #3: Yes

2. Has the statistical analysis been performed appropriately and rigorously? 

Reviewer #2: No

Reviewer #3: Yes

3. Have the authors made all data underlying the findings in their manuscript fully available?

Reviewer #2: Yes

Reviewer #3: Yes

4. Is the manuscript presented in an intelligible fashion and written in standard English?

Reviewer #2: No

Reviewer #3: No

5. Review Comments to the Author

Reviewer #2: The manuscript authored by Nakashima and coworkers describes in vitro findings on the effect of collagen and the chemokine CXCL12 on platelet aggregation and release of growth factor PDGF-AB, cytokine sCD40L and phosphorylated HSP27. The authors found that primarily the combination of collagen and CXCL12 stimulate the release of the determined compounds via activation of p38-MAPK. The authors claim that the described mechanisms may contribute to vascular inflammation and atherogenesis.

The hypothesis that the chemokine CXCL12 may be a prominent inflammatory mediator that synergistically induce platelet release reaction upon collagen-induced platelet activation is clear. The synergistically effect of platelet activation via inflammatory mediators is a timely issue that deserves intensive research work. The major limitation of the present study is that at present it remains descriptive with limited exploration of the underlying mechanisms. The authors convincingly show that collagen plus CXCL12 induce activation of the MAPK p38 and release of PDGF-AB, cytokine sCD40L, and phosphorylated HSP27. However, it is not clear whether this is specific for collagen-stimulated platelets or also is found for platelets stimulated with other agonists such as ADP or thrombin. Further, the receptor for collagen in this context should be further defined. Is this a specific GPVI-mediated effect? Specific experiments with collagen-related peptide (CRP) or convulxin will help. Further, CXCL12 binds to both CXCR4 and CXCR7 receptors on platelets. The authors need to show which is the prominent receptor for this putative synergistic effect on platelet release reaction and aggregation. And antibody-based approach using neutralizing anti-CXCR4 or anti-CXCR7 antibodies might disclose this aspect.

Specific comments

1. Table 1 and Figure 1 is redundant.

2. It is not clear why the authors used 10ng/ml of CXCL12 and 0.1µg/ml collagen. A dose-response curve would be of interest to validate the results. Is the synergistic effect of collagen and CXCL12 only found in the described and tested concentrations?

3. Fig 2B to Fig 2D should be combined for sake of clarity.

4. The discussion should be revised in regard a clear pathophysiological hypothesis where the described findings may play a role.

Reviewer #3: The manuscript is general sheds light on an important aspect of platelet biology, one in which CXCL12 plays a critical role in synergizing with collagen. We do have some issues that need to be addressed.

1. The “English” in the manuscript requires a lot of work, in terms of grammar and typos.

2. Abstract implies CXCL12 acts via CXCR only, whereas introduction via CXCR4 and CXCR7. This appears contradictory.

3. Elaborate on why the sCD40L assay the co-stimulation lasted for 15 minutes, unlike many of the other experiments.

4. SB203580 did not completely block the co-stimulation effects, which suggests a role for another pathway. What is it? Can you answer that experimentally?

5. All of the Western blot data should be quantified.

6. Another relevant article not cited is by Karim et al, 2016 in BBA (PMID: 26628381). It should be cited.

6. PLOS authors have the option to publish the peer review history of their article (what does this mean?). If published, this will include your full peer review and any attached files.

Reviewer #2: No

Reviewer #3: No

---

## [Author Response · Author response to Decision Letter 0]

17 Sep 2020

September 17, 2020

Yong Jiang, Ph.D.

Academic Editor

PLoS One

Dear Dr. Jiang,

Manuscript Number: PONE-D-19-34102R1

 Thank you very much for your kind E-mail on April 1, 2020. We are most pleased to learn that our manuscript entitled "Synergistic effect of collagen and CXCL12 in the low doses on human platelet activation" by D. Nakashima, et al. is acceptable for the publication in PLoS One after the adequate revision.

 We appreciate you and the reviewers for the proper estimation and the detailed review of our manuscript. According to the reviewers’ comments, the manuscript has been thoroughly revised. Our responses and changes made from the original manuscript are listed in the accompanying paper.

 We hope that our revised manuscript will be satisfactory for the publication in the journal. Would you please inform me of the decision by E-mail [okkasugai@yahoo.co.jp]? 

 With very best wishes,

 Sincerely yours,

 Osamu Kozawa, M.D. & Ph.D.

Department of Pharmacology, 

Gifu University Graduate School of Medicine, 

Gifu 501-1194, 

JAPAN

 E-mail: okkasugai@yahoo.co.jp

---

## [Editor Report · Decision Letter 1]

9 Oct 2020

Synergistic effect of collagen and CXCL12 in the low doses on human platelet activation

PONE-D-19-34102R1

Dear Dr. Kozawa,

We’re pleased to inform you that your manuscript has been judged scientifically suitable for publication and will be formally accepted for publication once it meets all outstanding technical requirements.

Kind regards,

Yong Jiang, Ph.D.

Academic Editor

PLOS ONE

---

## [Editor Report · Acceptance letter]

13 Oct 2020

PONE-D-19-34102R1 

Synergistic effect of collagen and CXCL12 in the low doses on human platelet activation 

Dear Dr. Kozawa:

I'm pleased to inform you that your manuscript has been deemed suitable for publication in PLOS ONE. Congratulations! Your manuscript is now with our production department. 

Kind regards, 

on behalf of

Professor Yong Jiang 

Academic Editor

PLOS ONE